# Medical Students’ Perception of Psychotherapy and Predictors for Self-Utilization and Prospective Patient Referrals

**DOI:** 10.3390/bs13010017

**Published:** 2022-12-24

**Authors:** R. Michael Drury, Nicki Taylor, Cheryl Porter

**Affiliations:** College of Medicine, Florida State University, Tallahassee, FL 32306, USA

**Keywords:** medical student, stigma, therapy, mental health, psychological disorders

## Abstract

The study explored if willingness to seek psychotherapy or refer patients to therapy is predicted by either perception of its usefulness or stigma (public and self-stigma), and if there are any differences based on specific psychological disorders for this population. A cross-sectional study was conducted surveying medical students enrolled at a southeastern university during spring 2022. These medical students completed the Mental Help Seeking Attitudes Scale (MHSAS), revised Self-Stigma of Seeking Help Scale (SSOSH-7), Stigma Scale for Receiving Psychological Help (SSRPH), in addition to vignette-based items assessing the likelihood they would seek therapy treatment and refer a patient for therapy based on two specific psychological disorders. The survey was completed by 106 medical students out of 495 current students (21.4% response rate). The data reveal that the greater the public stigma endorsed, the less likely medical students would be willing to seek therapy for panic disorder; however, the greater the self-stigma, the less likely they would seek therapy for depression. This study found differences in willingness to utilize therapy based on stigma-related attitudes and specific disorders, but not for referring patients.

## 1. Introduction

It is well documented that medical students’ mental health and stigma-related attitudes are concerns within this population [1,2]. Indeed, medical students endorse higher rates of emotional disorders and psychological distress compared with other postsecondary graduate students [3]. The majority of medical students may not seek out treatment for burnout or emotional difficulties [4], and recent research revealed that when medical students identify a need for mental health support, a large proportion do not use available services [5]. Regarding mental health support services, medication is often an important intervention, particularly since some studies have found combination treatment of pharmacotherapy and psychotherapy, namely Interpersonal Psychotherapy (IPT), to be more efficacious than IPT alone for depression [6], as an example. Given that there would likely be varying attitudes towards both modalities, and combining them would make it difficult to disentangle perception of each modality, this study focuses primarily on psychotherapy as mental health support. In addition, studies have explored medical students’ attitudes toward mental health and found that medical students with depression reported stigma-related attitudes regarding depression more often than medical students without depression [1,2]. These findings regarding accessing services or perception of mental health issues also lead to concerns as they enter the workforce as providers, particularly given that stress and emotional challenges do not dissipate as they complete their medical education programs. To this point, approximately one in 15 physicians in the United States have suicidal thoughts within a one-year period, which is a higher prevalence than other workers surveyed in the United States during the same time frame [7]. It is imperative to continue formulating precise ways to better serve this population in order to promote well-being and self-care during their medical school experience, which can create a stronger foundation as they enter the medical field as practitioners.

Stigma related to mental health is broadly considered a negative perception associated with mental illness or seeking psychological services; specifically, public stigma describes a negative perception by society. In addition, self-stigma refers to the individual’s negative, often unacceptable, view of themselves if they were to seek help, which occurs when internalizing negative stereotypes [8,9,10]. Previous work by Vogel et al. [9] revealed that public stigma influences self-stigma, which can predict one’s attitudes toward therapy and, consequently, their willingness to engage in counseling services. While medical students generally endorse positive attitudes towards therapy, particularly if they have participated in treatment in the past, they are less likely to seek out this support themselves or refer patients for treatment if they believe the stigma to be greater than the benefits [11]. Fundamentally, understanding the threshold where students would consider benefits of psychotherapy to outweigh the possible stigma can create a framework in which to respond to misconceptions or provide appropriate education.

Stigma-related attitudes are not solely a barrier to treatment for the medical student, but can have future implications for their development as practitioners and the impact on health care treatment. For example, Corrigan et al. [12] found that providers who endorsed stigma-related attitudes were less likely to think patients with mental illness would adhere to a treatment, which then impacted their clinical decisions. In this regard, it is necessary to better understand the impact stigma has on seeking treatment for this specific population, in addition to exploring how stigma may interfere with referring patients to therapy. Few studies have examined the impact of mental health stigma on likelihood to seek therapy [5,11,13], and there have not been any publications discovered by these authors that specifically explored if stigma, either public stigma or self-stigma, would have different implications for seeking therapy and referring patients based on specific psychological disorders. Given the prevalence of depression among medical students, it is important to expand the research to also include conditions that medical students are likely to experience or encounter in the medical field. For example, there is a financial and time-consuming toll that panic disorder can have on patients and medical systems when undiagnosed and/or untreated [14]. That being said, approximately one-third of medical specialists knew that cognitive-behavioral therapy (CBT) is an efficacious treatment for panic disorder and that this specific therapy has been recommended as an initial consideration for a non-medical intervention by the American Psychiatric Association guidelines [15,16]. Moreover, only 6% of those medical specialists surveyed knew what the treatment entailed. While it is not expected that medical students or providers be knowledgeable about numerous therapeutic approaches, this deficit in awareness of an evidence-based psychological treatment may be an additional barrier when considering referring patients for therapy. 

In addition to stigma-related attitudes, some recent studies have highlighted specific barriers to seeking help, such as accessibility (e.g., lack of time) [5]. Research has also begun to focus on practical ways to reduce stigma and increase openness to consider therapy. One such study revealed that when experienced physicians engaged in self-disclosure regarding their history of mental health issues and treatment, it had a positive impact on medical students’ view of psychiatry and individuals with mental illness [17]. To date, there have been efforts to establish anti-stigma curricula through various methodologies and a majority of approaches studied have resulted in an improvement with stigma and mental health literacy [18]. Reducing stigma and creating more openness to utilize therapy is important since providing evidence-based psychological treatment for medical students has resulted in significant improvements in terms of distress, depression and anxiety [19]. 

This study extends previous research involving medical students and stigma-related attitudes, but more specifically, explores predictors for seeking therapy or referring patients under distinct contexts. The objectives include determining: (1) if willingness to seek support is predicted by perception of therapy usefulness; (2) if willingness to refer patients for psychotherapy is predicted by perception of therapy usefulness; (3) if a measure of self-stigma predicts willingness to seek support based on two specific situations involving a diagnosable psychological disorder; (4) if a measure of public-stigma predicts willingness to refer a patient in either of two specific situations involving a diagnosable psychological disorder. To explore these objectives, medical students completed a survey consisting of validated measures assessing attitudes toward seeking mental health support in addition to stigma (public and self), and a vignette-based situation in which two psychological disorders were presented and respondents were asked about their likelihood to seek support themselves or to refer a patient under those conditions. 

## 2. Materials and Methods

This cross-sectional study consisted of administering an online survey via Qualtrics to all currently enrolled medical students at one southeastern state university in January and February of 2022. Students received the introductory email with a link to complete the survey, in addition to two reminder emails disseminated at one-week intervals. Participation was completely voluntary and anonymous; the students completed an informed consent document and did not receive any compensation for completing the survey. The study was approved by the university’s institutional review board and consent to participate in the research was collected via Qualtrics. All participant data was analyzed with SPSS Statistics Version 26.

In addition to basic demographic questions, the survey included the Mental Help Seeking Attitudes Scale (MHSAS), a nine-item tool created to measure perceived usefulness of seeking mental health support, which has exhibited strong internal consistency (α = 0.92) and test–retest reliability [20]. To measure and compare stigma-related attitudes towards mental health service utilization, the survey also consisted of the revised Self-Stigma of Seeking Help Scale (SSOSH-7) and the Stigma Scale for Receiving Psychological Help (SSRPH), both of which have demonstrated good psychometric properties. The SSOSH-7 is a seven-item scale designed to assess self-stigma, referring to a negative perception of oneself as it applies to utilizing therapy, while the SSRPH consists of five questions addressing perception of societal views pertaining to seeking mental health services (i.e., public stigma). 

Given the breadth of psychological disorders that could be incorporated into this study, it is important to include Major Depressive Disorder, since it is prevalent among medical students and the broader population, and has been used in previous research regarding stigma by medical students [2]. In addition, Panic Disorder was chosen as an anxiety disorder given the likelihood that individuals experiencing the significant physiological symptoms and experiences associated with panic will first go to an emergency room, urgent care, or other medical setting. Therefore, medical professionals will likely encounter depression and panic across many medical environments, which provides a plausible diagnostic situation to explore willingness to refer such patients and if they are willing to self-initiate psychotherapy under the same context. Two vignette-based items included a description of a situation that meets criteria for Panic Disorder and Major Depressive Disorder, respectively, and asks respondents to endorse the likelihood that under those circumstances they would seek treatment and refer a patient for therapy. The symptomatology presented on these items was agreed upon by the three authors as fulfilling criteria for the disorders and meeting a threshold in which seeking therapeutic support would be a reasonable and recommended option. These items were rated on a Likert scale (1 = Very Unlikely … 5 = Very Likely) in terms of willingness to self-utilize or refer for psychotherapy. The items for Panic Disorder and Major Depressive Disorder are shown in Figure 1a,b, respectively. The survey also included items that explored if respondents have previously engaged in therapy and if they considered it to be favorable or unfavorable. While the COVID-19 pandemic has negatively impacted the mental health of medical students, particularly when there was a transition to digital learning only [21], this study focused on the likelihood to seek out or refer for treatment under the context of two separate psychological disorders rather than students’ current mental health issues or the impact of COVID-19.

Linear regression analyses were conducted to explore if type of stigma, perception of therapy usefulness, or other variables (e.g., year in program, age, race, gender) would predict likelihood to either utilize therapy or refer patients under each psychological disorder context. An additional paired *t*-test statistic was conducted to explore if there is a difference between mean scores generated on both the self-stigma (SSOSH-7) and public stigma (SSRPH) scales.

## 3. Results

A total of 495 medical students were invited to participate in this study; 106 students completed the survey, resulting in a 21.4% response rate. Of all respondents, about 26% were in their first year of medical school, 35% were in their second year, while 19% and 20% were in years three and four, respectively. The vast majority of respondents were under the age of 30 (91.5%) and identified as female (70.8%). About 26% of respondents identified as male and 2.8% identified as either transgender or non-binary. Most respondents were White (66%), with 12.3% Hispanic/Latino, 9.4% Asian or Pacific Islander, 6.6% Black/African American, 2.8% Middle Eastern, and the remaining respondents identified as Other (2.8%).

The findings revealed there is a small positive association between inclination to seek therapy for depression and perception of therapy usefulness (*B =* 0.232, *p* = 0.023); however, no other variables predicted willingness to seek treatment or refer patients based on perceived usefulness of therapy (Objectives 1 and 2). The data also revealed that there is a difference in regard to type of stigma and specific psychological disorders predicting likelihood to seek therapy (Objectives 3 and 4). As illustrated in Table 1, linear regression models revealed that public stigma is negatively associated with student willingness to seek psychotherapy when considering panic disorder (*B* = −0.549, *p* = 0.001), after controlling for all the other variables. In addition, an increase in self-stigma implies there is a decrease in student willingness to seek psychotherapy under the major depressive disorder context (*B* = −0.425, *p* = 0.005), after holding all other variables constant. There were no significant findings with respect to variables predicting willingness to refer a patient. Additionally, other findings with respect to age, year in program, race, and gender were not significant, and thus, are not reported in Table 1. Lastly, there was a stronger endorsement of public stigma (*M* = 2.46, SD = 0.72) compared to self-stigma (*M* = 1.82, SD = 0.72) which is significant, *t* (109) = −9.55, *p* < 0.01. 

Nearly all medical students (97%) either agreed or strongly agreed that better accessibility would increase their willingness to seek support. Second, a majority of students (81%) also indicated that if peers and colleagues shared about their experiences with therapy, it would increase their willingness to consider seeking this support. Next, 72% of students agreed that learning more about the efficacy/effectiveness data regarding specific treatments would have a positive impact on their willingness to attend therapy, followed by promotion of mental health by faculty (64%). Of note, 84% of respondents indicated they have engaged in psychotherapy in the past. When broadly asked about previous therapy experiences, approximately 69% of those surveyed indicated it was “favorable”, 12% endorsed “neutral,” 4% indicated it was “unfavorable,” and the remaining 15% selected “N/A.” 

## 4. Discussion

An important finding is that while there were no identified associations for whether or not a medical student would consider referring a patient to therapy for symptoms of panic or depression, there were predictors for the possibility they would seek help themselves under the same conditions. The results revealed that public stigma predicted willingness to seek treatment for panic disorder; specifically, the greater the public stigma that was endorsed, the less likely the student would be willing to seek treatment for this anxiety disorder. Additionally, public stigma did not predict willingness to seek therapy for depression, but rather, there was a negative association with self-stigma in that the greater the self-stigma, the lesser likelihood to consider seeking therapy for oneself for depression. In this regard, medical students throughout their training are acknowledging they would be less likely to seek treatment for depressive symptoms with higher self-stigma, and less willing to seek therapy for panic symptoms with greater public-stigma, although they would be willing to refer patients for treatment under the same circumstances. The concept of delaying or declining treatment could become a larger concern as they may continue to delay treatment for symptoms throughout residency and beyond. 

In general, medical students endorsed public stigma more strongly than self-stigma; however as described above, stigma-related attitudes can vary dependent upon the psychological disorder or symptomatology, potentially interfering with help-seeking behaviors differently. To explore ways to navigate these obstacles to utilizing services, this study also explored if there were certain actionable items that would attenuate the barriers created by stigma. While this was not intended to be an exhaustive list, it continues to demonstrate that improved accessibility to services; the normalization of mental health support by peers, colleagues and faculty; and providing information about evidence-based treatments are potential interventions that can increase openness to utilizing therapy. Medical education programs should consider ways in which to incorporate these concepts into direct interventions and as part of the culture to promote better overall well-being. The findings of this study further illustrate the need to begin these interventions as early as the first year of medical school, as these results indicate medical students may not seek treatment for their own clinical symptoms based on stigma-related attitudes throughout their program.

Overall, when controlling for all other variables, there were no predictors for a medical student to consider referring a patient for either psychological disorder. It is important to acknowledge that while there may be fewer barriers pertaining to stigma when considering referring patients to therapy, there remain some potential challenges for medical students seeking their own therapeutic support. While this study was limited to symptoms of panic and depression, it would be helpful to explore how stigma-related attitudes may impact help-seeking or referrals of other psychological disorders, including other anxiety disorders, mood disorders, psychotic disorders, and/or personality disorders, to name a few. It would be recommended to examine the relationship between these disorders, stigma, and seeking therapeutic support or referring patients. In addition, there would be benefit in identifying perception of pharmacotherapy and psychotherapy given distinct contexts, as some individuals may identify different costs or benefits for medications compared to psychotherapy. The data obtained during this study has provided a foundation for future researchers to further examine the relationship of stigma-related attitudes based on specific symptomatology for medical students, in addition to specific stigma-reducing concepts and interventions that can engender more openness and accessibility to mental health services during medical students’ education and beyond.

Limitations to this study include the small sample size and reduced external validity based on the sample being located at one southeastern university medical school. Additionally, the 21.4% response rate is lower than anticipated, and females were slightly overrepresented in the sample (70.8%) compared to the student population (58.4%). There was, however, a relatively proportional distribution of respondents across all four years of the program. This study also explored intentions of utilizing or referring to therapy given a hypothetical situation, rather than actual choices to a current situation. 

## 5. Conclusions

Stigma-related attitudes can predict the willingness to seek mental health services for medical students depending on the specific type of psychological disorder (i.e., Panic Disorder, Major Depressive Disorder). Regardless of which type of stigma endorsed, medical students tend to view their own help-seeking needs differently than they would for a patient, and at times, may be less willing to seek therapeutic support for themselves. Future studies should further explore the differences in specific psychological disorders and the implications those can have on help-seeking behaviors. 

## Figures and Tables

**Figure 1 behavsci-13-00017-f001:**
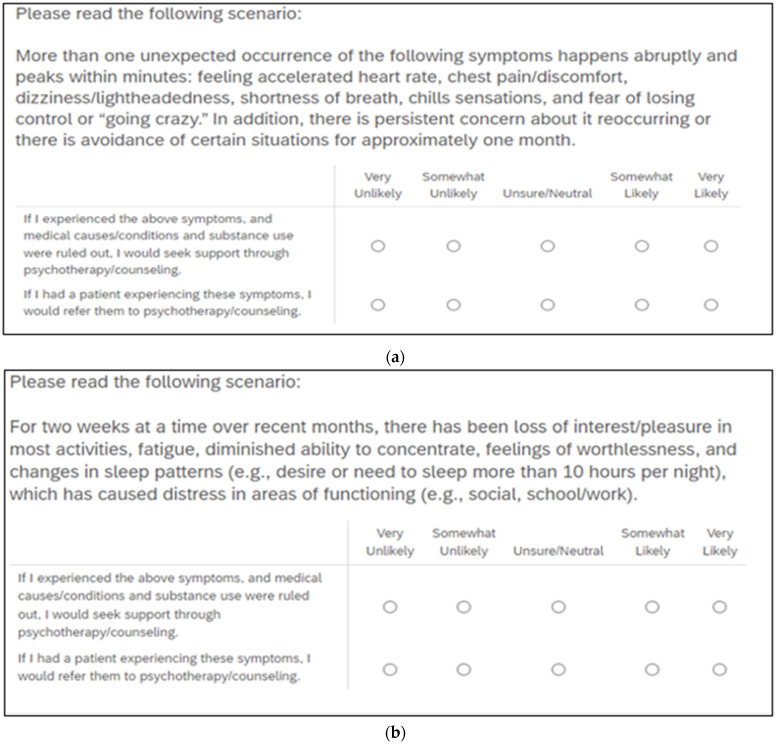
(**a**): Panic Disorder. (**b**): Major Depressive Disorder.

**Table 1 behavsci-13-00017-t001:** Linear regression analysis results for predictors of perceived likelihood to seek therapy with Panic Disorder and Major Depressive Disorder.

Predictor Variables	*B*	Standard Error	*p*
Panic Disorder ^a^			
Usefulness of therapy	0.116	0.112	0.303
Self-Stigma	−0.190	0.164	0.249
Public Stigma	−0.549	0.164	0.001 *
Major Depressive Disorder ^b^			
Usefulness of therapy	0.232	0.101	0.023 *
Self-Stigma	−0.425	0.148	0.005 *
Public Stigma	−0.175	0.148	0.241

R squared = 0.29 ^a^, 0.39 ^b^; * *p* < 0.05; *B* = unstandardized coefficient.

## Data Availability

Data is not currently publicly available; however, it is available upon request.

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
