# Peer review of "Medical Students’ Perception of Psychotherapy and Predictors for Self-Utilization and Prospective Patient Referrals"

_behavsci, 2022, doi:10.3390/bs13010017_

Round 1
Reviewer 1 Report
This paper presents an study for assessing medical students' attitudes for seeking and referring patients to psychotherapy. The manuscript is clear, understandable, well-written, and interesting for a wide audience. The study has several limitations identified by the authors in the discussion section. Some points should be fixed for strenghting the manuscript:
1. Table 1 does not contain the demographic characteristics of the participants as line 152 states.
2. There are some results missing. In lines 145-149, the authors declare that survey includes variables related to year in program, age, race, gender, etc.; it would be interesting state if such variables predict likelihood to either utilize therapy or refer patients. Even they do not, showing results would be interesting. Furthermore, it is also claimed that mean scores were compared by means of t-test statistics. Those results should be included too.
3. Analyzing the relation between having engaged in psychoterapy in the past and the willingness to refer patients or seek mental therapy would be interesting.
4. Line 151: "T A total" > "A total"
5. There would be worthy to include a Conclusion section to remark which are the conclusions extracted from the results, and if they may identify some actions ahead to reduce the stigma of mental issues among medical students.
Author Response
Please see the attached file "Responses to Reviewer 1." Thank you very much for your feedback and time.

Reviewer 2 Report
Thank you for the opportunity to review the paper entitled “Medical students’ perception of psychotherapy and predictors for self-utilization and prospective patient referrals.”
Thank you for the current opportunity for the peer-review.
However, there are several points which should be considered to improve the quality of the study.
I hope it could be a tip to consider to refine the article.
1. Abstract
“Barriers related to stigma” sounds a little bit ambiguous. Please clarify it in accord to the study.
2. Introduction
The phrase “stigma attitudes toward mental health” does not seem reasonable.
3. Introduction
Please clarify the definition of “self-stigma” in this study. Originally, it is the conception among those who experiences negative events such as mental illness.
4. Introduction & Study aim
I did not understand why the authors focused on only psychotherapy, instead of comprehensive psychiatric treatment. As both of pharmacological treatment and psychotherapy are provided together for many patients with mental illness generally, the explanation in this section did not seem enough. Negative attitudes towards pharmacological treatment and psychotherapy should not be the same. Therefore, the authors may want to explain about it.
5. Introduction & Study aim
I’m not sure whether it is reasonable to ask “to seek support” and “to refer a patient” together, as these are totally different. To seek support by themselves should be more closely related to self-stigma.
6. Introduction & Study aim
Regarding study aim, 3) if there are differences regarding public and self-stigma for the medical student population, the aim and analysis did not sound adequate and coherent. The expression of the aim #3 means to compare the scores between public stigma and self-stigma, but it does not seem make sense.
7. Materials and Methods
Please describe why the authors chose panic disorder and major depressive disorder in this study.
There are various mental illnesses which show the symptom of anxiety. Besides, many past studies showed that the stigma towards people with depressive disorder, and people with schizophrenia are quite different.
8. Materials and Methods
“…this study focused on the likelihood to seek out or 138 refer for treatment under specific conditions”
Please describe the “under specific conditions” in detail.
9. Materials and Methods
As mentioned in #6, it does not seem appropriate to simply compare the mean scores on two scales,. Besides, in Result section, there is not any result about it.
10. Results
The results were not accordant with how to analyze the data in Methods section.
Comprehensive review of the study aim and to analyze the data again would be needed.
11. Results
More concrete and accurate description about figures is needed. For example, what does “B” mean?
If it is standardized beta, it should be written as such.
12. Discussion
The discussion seems too short. More concrete discussion based on the finding of the study may give more implication for further study.
Author Response
Please see the attached file, "Responses to Reviewer 2." Thank you very much for your feedback and time.

Round 2
Reviewer 2 Report
Thank you for the opportunity to review the revised paper.
While it was revised, some revisions sound more superficial than rigorous.
For example, Points such as #3, #5 #7 from Reviewer #2 were not resolved or explained enough.
Besides, discussion section is still brief and conclusion section seems too lengthy.
More comprehensive and profound refinement based on the points that reviewers suggested before would improve this article splendidly.
Round 3
Reviewer 2 Report
Thank you for the opportunity to review the revised paper (v.3). I confirmed that several points were improved and it is now more understandable.